# Small Demyelination of the Cortex May Be a Potential Marker for the Right-to-Left Shunt of the Heart

**DOI:** 10.3390/brainsci12070884

**Published:** 2022-07-05

**Authors:** Junyan Huo, Mengxia Wan, Nan Li, Juan Wang, Xiao Cai, Dongsheng Fan, Yu Fu

**Affiliations:** 1Department of Neurology, Peking University Third Hospital, Beijing 100191, China; huojunyan1329@163.com (J.H.); wanmengxia12@163.com (M.W.); futingwang@sina.com (J.W.); caixiao519@163.com (X.C.); 2Research Center of Clinical Epidemiology, Peking University Third Hospital, Beijing 100191, China; linan917@163.com

**Keywords:** migraine, white matter lesions, right-to-left shunts

## Abstract

Migraine is a common clinical primary headache with unclear aetiology. In recent years, studies have shown that migraine is related to right-to-left shunts (RLS), and some patients with migraine have white matter lesions. However, the relationship among the three is unclear. To explore the characteristics of white matter lesions (WMLs) in migraine patients with right-to-left shunts and to predict the presence of right-to-left shunts through magnetic resonance imaging (MRI) characteristics in patients with migraine, we conducted a retrospective study. We enrolled 214 patients who were diagnosed with migraines in an outpatient clinic from January 2019 to December 2021. All of them had completed contrast transcranial Doppler ultrasound (cTCD) and magnetic resonance imaging (MRI) examination. Through the inclusion and exclusion criteria, 201 patients were finally included. The patients were grouped according to the presence of WMLs and were compared by age, sex, hypertension, diabetes, RLS, and other characteristic data. We observed the MRI fluid attenuation inversion recovery sequence (FLAIR) image and compared the differences in WMLs between the RLS-positive group and the RLS-negative group. There were 71 cases and 130 cases of migraine with and without WMLs, respectively. A statistically significant difference in near-cortical WMLs with RLS in migraine patients was observed (*p* = 0.007). Logistic regression analysis was adjusted by age, sex, duration of migraine, and severity. Migraine with aura and family history identified the RLS status as the sole determinant for the presence of near-cortical WMLs (OR = 2.69; 95%CI 1.386–5.219; *p* = 0.003). Near-cortical white matter lesions in migraine patients are related to RLS, especially in the blood supply area of the anterior cerebral artery. This small demyelination of the near-cortical WMLs may be a potential marker for the right-to-left shunt of the heart. Transcranial Doppler ultrasonography may help finding more RLS in migraineurs with near-cortical WMLs.

## 1. Introduction

Migraine is a chronic neurological disease that is characterized by moderate or severe headaches. Typical symptoms include photophobia, phonophobia, skin allodynia, and gastrointestinal symptoms, such as nausea and vomiting [1]. The prevalence of migraine in the Chinese adult population diagnosed with primary headaches is approximately 9.3%, and it is listed as one of the seven major causes of disability by the World Health Organization [2]. In the past, it was believed that no imaging manifestations could be detected in migraine patients. However, with the development of magnetic resonance imaging (MRI), white matter lesions (WMLs) have been found in more and more migraine patients, usually manifesting as multiple and small punctate lesions in the white matter [3]. The relationship between migraine, the patent foramen ovale, and white matter lesions is not clear. Studies have shown that migraine and the patent foramen ovale are related with each other [4]. In some migraine patients, the headache can disappear or be relieved after the closure of the foramen ovale [5]. However, percutaneous closure of the PFO in the treatment of migraine is still controversial [6]. In addition, white matter lesions in migraine patients increase the risk of syncope and orthostatic intolerance [7]. In patients with PFO, the blood is shunted from right to left, which is one of the causes of cryptogenic embolism. Although the mechanism of WML has not been fully elucidated, the presence of white matter lesions may lead to uncertainty for physicians and anxiety for patients. Therefore, it is of great clinical significance to explore the white matter lesion characteristics of migraine patients with a right-to-left shunt and to predict the existence of a right-to-left shunt using the MRI characteristics of migraine patients. In this study, we observed the abnormal changes of white matter in migraine patients and discussed whether RLS increased the risk of specific WMLs in migraine patients, so as to provide references for clinical diagnosis and treatment in the future.

## 2. Materials and Methods

### 2.1. Study Population

In this retrospective study, a total of 214 migraine patients who were admitted to the Neurology Clinic of Peking University Third Hospital from January 2019 to December 2021 were included, and all of them had completed a contrast transcranial Doppler ultrasound (cTCD) and magnetic resonance imaging (MRI) examination. Considering that white matter lesions caused by factors such as arteriosclerosis in those older than 50 years of age would interfere with our results, we set the age of the study population to be less than 50 years old. Through the inclusion and exclusion criteria, 201 migraine patients were finally enrolled. The inclusion criteria were as follows: (1) met the diagnostic criteria of ICHD-3; (2) aged 18–50 years; (3) received contrast transcranial Doppler ultrasound (cTCD) and magnetic resonance imaging (MRI) examination. The exclusion criteria were as follows: (1) other types of primary headaches; (2) history of cardiovascular and cerebrovascular diseases; (3) intracranial organic diseases; (4) demyelinating disease and hereditary white matter disease; (5) tumour; (6) contraindications on MRI examination; (7) diagnosis of vascular stenosis (vascular stenosis >50%). We included patients who had either transesophageal echocardiography or agitated saline contrast echocardiography. This research was approved by the Ethics Committee of Peking University Third Hospital. Informed consent was obtained. All methods were performed in accordance with the relevant guidelines and regulations.

### 2.2. Clinical Data Collection

Data collected included age, sex, past history, duration of migraine, the presence of aura, and the family history of migraine. In addition, the severity of headaches was scored on a visual analogue scale (VAS).

### 2.3. RLS Assessment

Using a transcranial Doppler ultrasound diagnostic apparatus and a 2 MHz probe, the subject was placed in the supine position. The neurologist created a single-channel multi-depth mixture of 8 mL normal saline, 1 mL air, and 1 mL autologous venous blood. The saline solution was activated and injected quickly into the median vein of the right elbow, and changes in the blood-flow spectrum signal of the bilateral middle cerebral arteries in the resting state and after the Valsalva manoeuver were detected. Microbubbles (MBs) in the resting state or after the Valsalva manoeuver indicated positivity for RLS; negativity for RLS was indicated if MBs were not detected. The cTCD test positive standard was as follows: MBs appearing within 10 s were defined as positive and were classified according to the number of MBs. The classifications were Class 0 (no MBs), Class I (1–10 MBs), Class II (more than 10 MBs but no rain curtain), and Class III (forming a rain curtain).

### 2.4. MRI Examination

Using a 3.0 T superconducting magnetic resonance scanner, the subjects were asked to lie on their backs, relax, close their eyes, and stay awake during data collection. T1WI, T2WI, and fluid-attenuated inversion recovery (FLAIR) images were collected, with a layer thickness of 5 mm and a layer spacing of 6 mm.

### 2.5. Image Post-Processing Analysis and WML Evaluation

WMLs were evaluated by 2 experienced neurologists who used the image processing workstation to assess the WMLs of T2 FLAIR images in the inclusion group. Data with imaging artefacts were not included in this study. The characteristics of WMLs were defined as follows: high signal on T2WI and T2 FLAIR sequences and equal or low signal on T1WI sequences. Periventricular WMHs (pvWMHs) were assessed in three regions (frontal and posterior horns and bands) [8]. Deep WMHs (dWMHs) were located in deep white-matter tracts and were not attached to lateral ventricle lesions [9]. Near-cortical WMHs on FLAIR images were defined as small areas of hyperintensities in the cortex and cortico-subcortical area [10]. The white matter lesions on the MRI were punctate white matter lesions, which are not applicable to the Fazekas scale. To ensure good consistency in this study, when the two evaluators disagreed, a consensus was reached through discussion. We selected typical images for Figure 1.

### 2.6. Statistical Analysis

SPSS 24 was used for data processing. Count data were expressed by the rate, and comparison between groups was expressed by the chi-square test. For measurement data, those with normal distribution after inspection were expressed by the mean ± standard deviation (x ± s), and comparison between the groups was expressed by two independent sample t-tests. If the measurement data did not conform to the normal distribution, the data were represented by the median and quartile (M (Q25, Q75)), and comparison between the groups was carried out using two independent sample nonparametric tests. The presence of a linear trend between groups was assessed through linear association. When there was a correlation between the two ordinal categorical variables, we used Kendall’s tau b correlation analysis; multivariate logistic regression was used for the analysis of independent influencing factors of WMLs in migraine patients. A *p* value < 0.05 indicated a statistically significant difference.

## 3. Results

### 3.1. Comparison between the WML-Positive and WML-Negative Patients

According to the inclusion and exclusion criteria, a total of 201 migraine patients were enrolled in this study. The mean age of the patients was 32.66 ± 7.51 years, and the mean disease duration was 7.61 ± 6.02 years. The sex ratio of female to male was 1.39:1. As shown in Table 1, 71 patients (35.3%) had WMLs, and 130 patients (64.7%) did not have WMLs; 39 patients in the WML group had RLS (54.9%), and 56 patients (43.1%) in the group without WMLs had RLS. Patients with RLS had a higher frequency of WMLs than did those without RLS. There were no significant differences in age, sex distribution, duration of migraine, severity of migraine, migraine with aura, or RLS between the groups, and there was not a linear correlation trend between the WMLs and RLS grades.

### 3.2. Comparison between the Near-Cortical WML-Positive and WML-Negative Patients

The patients were grouped according to whether or not there were near cortical white matter lesions (Table 2). We found 56 subjects with near-cortical WMLs and 145 subjects without near-cortical WMLs among the migraineurs in this study. The results showed the prevalence of RLS in the near-cortical WML-positive group was significantly higher than that those in the near-cortical WML-negative group (RLS, 62.5% vs. 41.4%, *p* = 0.007), and there was a linear correlation trend between near-cortical WMLs and RLS grade. We counted the number of near-cortical white matter lesions and the grade of RLS. As shown in Table 3, there was no correlation between the number of white matter lesions and RLS grade. Kendall’s tau b = 0.115, *p* = 0.064.

### 3.3. Comparison of White Matter Lesions between the RLS-Positive Group and the RLS-Negative Group

As shown in Table 4, among migraineurs, patients with RLS had a higher frequency of near-cortical WMLs than did those without. There was a statistically significant difference in the distribution of WMLs in the anterior cerebral artery between the RLS-positive and RLS-negative group (*p* = 0.008). We carried out a univariate logistic regression analysis and multivariate logistic regression analysis, respectively. The multivariate analysis was conducted to adjust for age, sex, and migraines with aura that may have influenced the WMLs’ prevalence. The multivariate logistic regression analysis indicated that the presence of RLS was independently associated with the distribution vascular zone of WMLs (*p* = 0.011; OR = 2.53; 95% CI 1.238–5.168). As shown in Figure 2, the sensitivity of the near-cortical white matter lesions in predicting RLS was 62.5%, and the specificity was 37.5%, The area under the ROC curve was 0.672, which had good predictive value. This small subcortical demyelination may be a potential marker for the right-to-left shunt of the heart.

### 3.4. Multivariate Logistic Regression Analysis of White Matter Lesions and Near-Cortical White Matter Lesions

A multivariate logistic regression analysis was conducted to adjust for age, sex, duration of migraine, presence of aura, and family history of migraine. The results indicated these factors were not associated with the presence of WMLs (Table 5). According to the multivariate logistic regression analysis, the presence of RLS was associated with the presence of near-cortical WMLs (*p* = 0.003; OR = 2.69; 95%CI 1.386–5.219).

## 4. Discussion

Migraine is a chronic neurological disease that is more common in women than in men, with a male-to-female ratio of 1:3. The onset of migraine usually starts in late childhood or early adolescence, and the incidence peaks in middle age [11]. Decades of headaches recur during the active cycle and may become chronic or more difficult to treat in some patients [12]. Migraine has been identified by the World Health Organization as one of the main causes of global disability, especially for individuals under 50 years of age [13]. The aetiology of migraine is unclear.

The foramen ovale is the physiological passage of the heart chamber during the foetal period. It is usually closed by the time an individual reaches the age of 2. PFO is caused by the failure of the fusion between the primary septum and the atrial septum. In the general population, the rate of PFO is approximately 15–35% [14]. PFO is the most common right-to-left shunt disease, accounting for approximately 95% of all right-to-left shunt diseases. When a patient has PFO, there will be a permanent shunt of blood from the right atrium to the left atrium or only when the right atrium pressure increases further, such as with the Valsalva manoeuver. RLS is the main cause of paradoxical embolism.

Studies have shown that some migraine patients show white matter lesions on an MRI. The main feature of WMLs is a high signal on T2-weighted or FLAIR images [15]. In general, hyperintensity on FLAIR or T2 sequences tends to be secondary to axonal degeneration, gliosis, and demyelination. There are two possible mechanisms for white matter lesions on MRI in migraine patients: the first hypothesis is that migraine causes vasoconstriction and ischaemia that damage axonal myelin sheaths. A second hypothesis is that migraine stimulates metabolic processes that damage myelin sheaths [16]. Although the aetiology of WMLs is unknown, recent studies have shown that WMLs may originate from ischaemia. In addition, haemodynamic changes may be related to white matter ischaemia [17,18].

PFO accounts for approximately 95% of all right-to-left shunt diseases. It occurs at birth, while white matter lesions are often acquired. Therefore, it is reasonable to think that RLS precedes white matter lesions. Our findings based on retrospective observation suggest a significant relationship between RLS and near-cortical WMLs in these patients with migraine. In migraine patients with RLS, WMLs are mostly distributed in the frontal and parietal lobes, which is in the blood supply area of the anterior cerebral artery. However, the exact mechanism is unclear. We speculate that white matter lesions near the cortex may be due to an embolic mechanism, since a right-to-left shunt is mostly a small tunnel. As the thoracic pressure changes, the emboli are mostly small emboli that pass intermittently. With the heart pumping 10% of the blood into the skull, microemboli are more likely to enter the distribution area of the anterior cerebral artery because the anterior cerebral artery runs straighter than other vessels and is distributed along the blood vessels to the most distal, more common in the frontal lobe. Some emboli autolyse without lesions, which may explain the absence of obvious clinical symptoms in many patients. Some of the larger emboli will remain in the arterioles, where the emboli block the microcirculation and lead to local hypoperfusion, thus producing subcortical white matter lesions. But since the microemboli are 1–3 mm, it is impossible to show how these microemboli get from the right atrium to the left atrium [19]. The source of microemboli is unknown. Previous studies have shown that many cryptogenic strokes may be caused by small emboli that spread from the legs to the right atrium. It is possible that the source of these microemboli has the same mechanism as that of cryptogenic stroke. We did not find a correlation between white matter lesions and aura, and the results were similar to those of previous studies [20,21]. Some observational studies have shown that migraine patients with aura have higher rates of PFO than the general population, but whether there is an association between migraine aura and right-to-left shunts is still controversial [22]. No difference in age, sex, hypertension, or diabetes in patients with or without RLS was found for these near-cortical WMLs. There was not a linear correlation trend between the WMLs and RLS grades. The maximum number of microbubbles (MBs) was taken as the estimate of the maximum degree of shunt, since the microbubbles do not necessarily form microemboli. A study showed that WMHs did not increase with the size of RLS [8]. A review showed that the sex ratio of females to males in the United States is about 3:1 [23], while a Chinese epidemiological study of migraine showed that the ratio is 2.15:1^2^, which can be attributed to racial differences. In our study, the sex ratio of females to males was 1.39:1, which was significantly lower than other studies. It may be that since our study was a single-centre study, there may be some bias in the registered outpatient patients.

Our study was consistent with a previous Korean study, which indicated the relationship between the presence of near-cortical WMLs and RLS in migraine patients. A total of 98 migraine patients in South Korea were recruited. FLAIR images showed that WMLs near the cortex were more common in those with right-to-left shunts compared with the rest [10]. In our study, we additional detected arterial subregions in which white matter lesions were located.

Some studies have reported opposite results. For instance, Park’s study in Korea showed that 89 of 242 migraine patients had right-to-left shunts, demonstrating that deep WMLs with low magnetic fields were independently related to the right-to-left shunts of migraine patients [20]. In 2018, a multi-centre study involving 334 migraine Chinese patients found that right-to-left shunts were not associated with WMLs in migraine patients [8]; however, in that study, only deep white matter lesions and periventricular white matter lesions were considered. Another study conducted in 139 children and adolescents with migraine in the United States showed that WMLs in migraine patients did not correlate to the existence of right-to-left shunts, the degree of right-to-left shunts, nor the subtype of migraine [24]. However, some of patients included in this study did not undergo an MRI examination. Overall, the different results of these studies may be related to the races, definitions and classification of WMLs, age, and in the magnetic resonance equipment, the setting parameters and research methods.

The migraine patients included in this study were compared by age, sex, past history, duration of migraine, presence of aura, and family history of migraine. It was preliminarily concluded that RLS was associated with near-cortical WMLs, especially in the areas of anterior cerebral artery supply. Currently, there is controversy regarding the treatment of migraine with PFO closure surgery, but the preventive effect of PFO closure surgery on cryptogenic embolism has been confirmed by large-scale experiments [25,26,27]. Therefore, for migraine patients with subcortical WMLs, cTCD can detect RLS, and occlusion treatment in advance can prevent the occurrence or recurrence of stroke in the future.

Although this study obtained some preliminary conclusions, there are still some shortcomings. First, the sample size was small, which may lead to bias. However, we found a phenomenon that migraine was combined with subcortical white matter lesions, suggesting the possibility of RLS. This finding is worthy of further confirmation. The second shortcoming was that the VAS scale was relatively simple and may have underestimated severity. It would have been better to use the MIDAS questionnaire or the HIT-6 scale.

## 5. Conclusions

This study found that RLS was associated with near-cortical WMLs in migraine patients and that the lesions were mostly located in the blood supply area of the anterior cerebral artery. There are still some shortcomings. For example, the sample size was small, and the source of the microemboli was unknown. It is worth expanding the sample further, since more research is needed to confirm our results and to investigate the clinical implications. This small demyelination of the subcortex may be a potential marker for the right-to-left shunt of the heart. Therefore, cTCD for migraine patients who present clinically with near-cortical WMLs will help to identify the cause and provide a basis for future treatment decisions.

## Figures and Tables

**Figure 1 brainsci-12-00884-f001:**
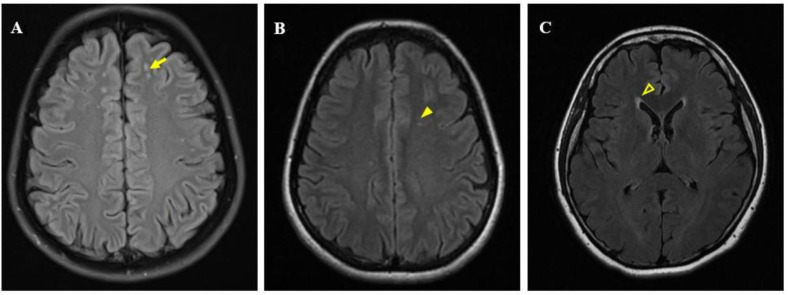
Fluid-attenuated inversion recovery images. (**A**) Near-cortical white matters (yellow arrow), (**B**) deep white matter lesions (yellow solid arrow), and (**C**) paraventricular matter lesions (yellow hollow arrow).

**Figure 2 brainsci-12-00884-f002:**
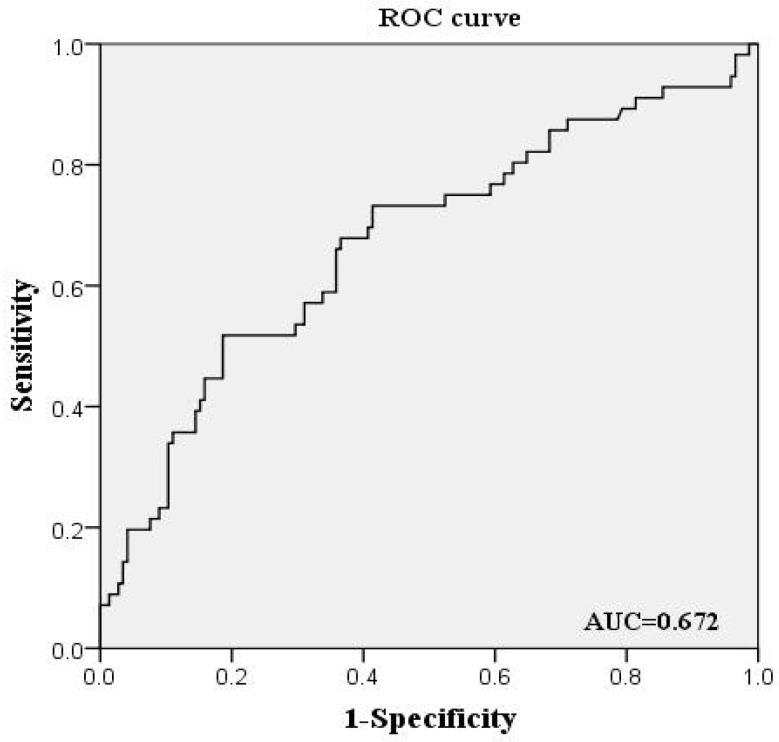
The ROC curve for the prediction of a right-to-left shunt in the near-cortical white matter lesions.

**Table 1 brainsci-12-00884-t001:** Comparison of characteristics between the WML-positive and WML-negative groups.

Clinical Data	WMLs(+) (*n* = 71 )	WMLs(−) (*n* = 130 )	*p*
Age (y)	33.56 ± 7.56	32.17 ± 7.46	0.197
Female, *n* (%)	40 (56.3)	77 (59.2)	0.691
Diabetes, *n* (%)	1 (1.4)	2 (1.5)	0.715 ^a^
Hypertension, *n* (%)	6 (8.5)	5 (3.8)	0.148
Duration of migraine, *n* (year)	5 (2, 11)	6.5 (3, 11.25)	0.243
Severity	4.94 ± 1.88	4.69 ± 1.92	0.776
Migraine with aura, *n* (%)	11 (15.5)	34 (26.2)	0.691
Family history of migraine, *n* (%)	13 (18.3)	30 (23.1)	0.431
RLS	39 (54.9)	56 (43.1)	0.168
RLS Grades			
0	32 (45.1)	74 (56.9)	0.210 ^b^
I	17 (23.9)	32 (24.6)	
II	8 (11.3)	10 (7.7)	
III	14 (19.7)	14 (10.8)	

Data are presented as the mean ± standard deviation, median (first quartile, third quartile), or *n* (%). Abbreviations: ^a^ Fisher’s test; ^b^ linear-by-linear association; RLS, right-to-left shunt; WMLs, white matter lesions.

**Table 2 brainsci-12-00884-t002:** Comparison of characteristics between the near-cortical WML-positive and WML-negative groups.

Clinical Data	Near-Cortical WMLs(+) (*n* = 56)	Near-Cortical WMLs(−) (*n* = 145)	*p*
Age (y)	34.11 ± 7.96	32.10 ± 7.28	0.068
Female, *n* (%)	33 (58.9)	84 (57.9)	0.898
Diabetes, *n* (%)	1 (1.78)	2 (1.37)	0.627 ^a^
Hypertension, *n* (%)	6 (10.7)	5 (3.4)	0.076 ^a^
Duration of migraine, *n*(year)	6 (2, 11)	6 (3, 11)	0.636
Severity	5.13 ± 1.91	4.65 ± 1.89	0.415
Migraine with aura, *n* (%)	10 (17.9)	24 (16.6)	0.825
Family history of migraine, *n* (%)	11 (19.6)	32 (22.1)	0.707
RLS, *n* (%)	35 (62.5)	60 (41.4)	0.007
RLS Grades			
0	21 (37.5)	85 (58.6)	0.001 ^b^
I	14 (25.0)	35 (24.1)	
II	8 (14.3)	10 (6.9)	
III	13 (23.2)	15 (10.4)	

Data are presented as the mean± standard deviation, median (first quartile, third quartile), or *n* (%). Abbreviations: ^a^ Fisher’s test; ^b^ linear-by-linear association; RLS, right-to-left shunt; WMLs, white matter lesions.

**Table 3 brainsci-12-00884-t003:** Correlation between the number of white matter lesions and RLS grade.

Number of WMLs	RLS Grade			
	0	I	II	III
0–5	97	46	14	25
6–10	5	2	3	3
11–15	3	0	1	0
16–20	1	1	0	0

Abbreviations: RLS, right-to-left shunt; WMLs, white matter lesions.

**Table 4 brainsci-12-00884-t004:** Comparison of white matter lesions between the RLS-positive group and the RLS-negative group.

			Univariate Logistic Regression		Multivariate Logistic Regression ^a^	
Lesion Location	RLS (+)	RLS (−)	OR (95%CI)	*p*-Value	OR (95%CI)	*p*-Value
Near-cortical, *n* (%)	35 (36.8)	21 (19.8)	2.36 (1.253, 4.451)	0.008	2.54 (1.326, 4.863)	0.005
Paraventricular, *n* (%)	13 (13.7)	22 (20.8)	0.61 (0.286, 1.282)	0.190	0.67 (0.310, 1.444)	0.307
Deep brain, *n* (%)	5 (5.3)	12 (11.3)	0.65 (0.257, 1.658)	0.370	0.68 (0.264, 1.732)	0.415
ACA blood supply area, *n* (%)	27 (28.4)	15 (14.2)	2.41 (1.190, 4.875)	0.015	2.53 (1.238, 5.168)	0.011
MCA blood supply area, *n* (%)	22 (23.2)	24 (22.6)	1.03 (0.533, 1.990)	0.931	1.07 (0.545, 2.100)	0.845
PCA blood supply area, *n* (%)	3 (3.2)	8 (7.5)	0.40 (0.103, 1.552)	0.185	0.45 (0.114, 1.795)	0.259

Abbreviations: ACA, anterior cerebral artery; MCA, middle cerebral artery; PCA, posterior cerebral artery; RLS, right-to-left shunt; ^a^ adjusted by age, sex, and migraine with aura.

**Table 5 brainsci-12-00884-t005:** Multivariate logistic regression analysis of white matter lesions and near-cortical white matter lesions.

	WMLs				Near-Cortical WMLs			
	Crude OR (95%CI)	*p*-Value	Adjusted OR (95%)	*p*-Value	Crude OR (95%CI)	*p*-Value	Adjusted OR (95%)	*p*-Value
Age	1.03 (0.986, 1.066)	0.209	1.03 (0.985, 1.070)	0.211	1.04 (0.994, 1.080)	0.091	1.04 (0.996, 1.092)	0.072
Sex	1.13 (0.627, 2.021)	0.691	1.19 (0.641, 2.207)	0.583	1.04 (0.557, 1.949)	0.163	1.07 (0.548, 2.107)	0.835
Duration of migraine	0.98 (0.934, 1.031)	0.464	0.97 (0.919, 1.027)	0.309	1.00 (0.954, 1.057)	0.870	0.99 (0.929, 1.045)	0.619
Severity	1.07 (0.921, 1.247)	0.370	1.14 (0.956, 1.363)	0.144	1.14 (0.970, 1.338)	0.113	1.18 (0.971, 1.422)	0.097
Migraine with aura	1.17 (0.535, 2.570)	0.691	1.21 (0.498, 2.920)	0.678	0.91 (0.405, 2.055)	0.825	0.95 (0.373, 2.429)	0.918
Family history	1.34 (0.647, 2.768)	0.432	1.40 (0.625, 3.117)	0.415	1.16 (0.538, 2.495)	0.707	1.43 (0.608, 3.371)	0.411
RLS	1.61 (0.900, 2.883)	0.109	1.80 (0.979, 3.277)	0.058	2.36 (1.253, 4.451)	0.008	2.69 (1.386, 5.219)	0.003

Abbreviations: RLS, right-to-left shunt; WMLs, white matter lesions.

## Data Availability

The data presented in this study are available upon request from the corresponding author. The data are not publicly available due to data management regulations in our hospital.

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
