# Peer review of "Small Demyelination of the Cortex May Be a Potential Marker for the Right-to-Left Shunt of the Heart"

_brainsci, 2022, doi:10.3390/brainsci12070884_

Round 1
Reviewer 1 Report
This is an intersting study showing possibler association between right-to-left shunt and white matter lesions among migraineurs. However, the manucript struggles with severe methodological issues.
First of all, the association of patent foramen ovale and migraine is not fully elucidated. Please see: https://www.acc.org/latest-in-cardiology/articles/2021/04/22/13/03/reevaluating-pfo-closure-for-migraine
The background of white matter lesions is also not clarified, but their clinical importance is doubtful: Palm-Meinders IH, Koppen H, Terwindt GM, Launer LJ, Konishi J, Moonen JM, Bakkers JT, Hofman PA, van Lew B, Middelkoop HA, van Buchem MA, Ferrari MD, Kruit MC. Structural brain changes in migraine. JAMA. 2012 Nov 14;308(18):1889-97.
The methodological part contains that vascular risk factors were excluded. Please clarify it. Furthermore, patinets with miagraine especially MA patients have higher Framingham scores than healthy ones. What about measuring blood lipids and screening for for prediabetes?
The most important part that cardiac evaluation was not carried at all, therefore the findings and the interpretation of results are most speculative than scientific.
Reviewer 2 Report
First of all, I would like to congratulate the authors for their work.
Migraine is a disease that, although globally well known by the general population and the scientific community, still has large gaps or unknowns regarding its possible etiology or interrelations with other diseases or pathologies. Therefore, it is not surprising that it continues to be the subject of many scientific studies and that much research is still needed in this regard.
Regarding the study presented by the authors, the possible relationship between migraine, white matter lesions and the right-left shunt of the heart is a topic that has been studied for some time by multiple authors and that seems to have relevance and scientific basis in terms of the study of migraine.
Although the topic is of scientific interest and the study design is correct, some aspects of the study should be reviewed or corrected for the quality of the study to be completely optimal.
Major revisions:
- Both in the last paragraph of the discussion (lines 261-263) and in the conclusions (lines 272-274) it is indicated that the results of the study suggest a possible causal relationship between the subcortical white matter lesions and the shunt right-to-left of heart.
Given that, according to the proposed study design, both variables were evaluated in the patients before their inclusion in the study, we do not know which of the two appeared first, or if there are other variables or factors that may influence the appearance or progression of these variables in patients with migraine. Therefore, neither causal possibility or causality between them can be affirmed. With the design and results of this study, one can only speak of an association between both variables.
Minor revisions:
- In section 2.2 (line 73) they indicate that to measure headache severity they used a visual analogue scale. Considering the complexity of symptoms involved in a disease such as migraine, the VAS is too basic a scale to measure this variable, because measuring pain alone to assess headache severity may miss many factors known to influence headache severity.
In this type of disease there are specific tools that measure more completely and exhaustively the severity and impact of headache pain on the patient's life, such as the MIDAS questionnaire or the HIT-6. Therefore, I consider that this issue should be mentioned in the limitations of the study, as it may influence the results of the study.
- In section 2.5 (lines 101 and 102), a series of acronyms are used for the first time in the document whose meaning is never indicated, such as WMH and dWMH. Although experts in the field know what these acronyms refer to, this is not reflected in the text, so that any reader knows their meaning.
- In section 3.4 (line 184) they indicate that "the presence of RLS was independently associated with the presence of near-cortical WMLs". It is likely that the error is a translation since the statistical analysis is correct, but it would be advisable to eliminate the word "independently" since it confuses the interpretation of the results and simply leave that they were associated.
- In the discussion section they highlight that, as has been shown in many studies in the last decade, migraine is a disease with a female predominance. Specifically, they indicate that the male-female ratio in migraine is 1:3 (line 192).
In the data of their study, this proportion is not met, observing a ratio of 1:1.39. I think it would be interesting to indicate what they believe may be due to this difference in proportions and to reflect it in the text.
- Also, in the discussion section (line 263-264) they indicate that a second preliminary conclusion of their study is that the source of microemboli is unknown. I think it is not correct to indicate that this is a preliminary conclusion of your study because it does not set itself the objective and does not evaluate the origin of microembolisms.
- In line with this topic, in the following lines (264-267) the possible origin of microembolisms is discussed, so if that second preliminary conclusion is eliminated, it makes little sense to leave this part of the text in that paragraph since it is not relates to what is spoken in it.
Given that said information is interesting in relation to the subject of study, and to prevent it from being lost, it would be advisable to move said information to another part of the text.
Round 2
Reviewer 1 Report
The authors have revised my concerns, therefore the manuscript meets the minimum criteria for publication.
Reviewer 2 Report
I congratulate the authors for their work and for the corrections made. These have helped to make the study clearer and more complete and to better appreciate the quality of the work carried out.
Despite the fact that with the corrections made, have almost completely resolved the comments on the text, I must insist that due to the type of study, the moment of evaluation of the variables and the statistical analysis carried out, it cannot be deduced a causal relationship between them, although, with the clarifications included in the text on PFO, a possible causal relationship between them can be suggested.
Therefore, I believe that the last lines of the conclusions in which it is indicated that: "Therefore, cTCD for migraine patients who present clinically with near-cortical WMLs will help to identify the cause and provide a basis for future treatment decisions" the part of "help to identify the cause and provide a basis for future treatment decisions" should be replaced by something similar that does not imply a causal relationship (since cannot be concluded from the study carried out but only suggested as a possibility).
It can be changed to something like "may help identify the cause and provide a basis for future treatment or decisions", which is very close to what the authors concluded in their study, but not without being a direct causal claim.